# The Impact of a Rater Training Program on the TGMD-3 Scoring Accuracy of Pre-Service Adapted Physical Education Teachers

**DOI:** 10.3390/children9060881

**Published:** 2022-06-13

**Authors:** Hyunjin Kwon, Hyokju Maeng

**Affiliations:** 1Department of Sport Culture, Dongguk University, Seoul 04620, Korea; clubkwon800@gmail.com; 2Department of Kinesiology and Health, Georgia State University, Atlanta, GA 30303, USA

**Keywords:** TGMD-3, adapted physical education, training, children, developmental disabilities

## Abstract

This study aimed to examine the impact of a TGMD-3 rater training program on pre-service adapted physical education (APE) teachers’ ability to score fundamental motor skills for children with developmental disabilities (DD). The training program consisted of a description of the TGMD-3 instrument and DD, as well as content focused on skill performance and correct scoring through systematic analysis of 13 FMS via the instrument. Three experts established the standard score through the TGMD-3 performance evaluation of a child with DD. Thirty-nine pre-service APE teachers in the experimental group and 30 pre-service teachers in the control group completed the pre- and post-test process in this study. There were significant impacts on the pre-service teachers’ ability to score the locomotor, ball skill subtests, and total skill scores (*p* < 0.05) compared to the control group. Specifically, the results of the three locomotor skills (running, horizontal jump, & slide) and three ball skills (two-hand strike, dribble, & kick) significantly improved in scoring accuracy. The results of this study can evidence that a TGMD-3 rater training program for pre-service APE teachers enhances the scoring accuracy of FMS among children with DD.

## 1. Introduction

In the U.S., about 1 in 6 (17%) children 3–17 years of age were diagnosed with developmental disabilities (DD) based on the 2009–2017 National Health Interview Survey (NHIS; Zablotsky et al., 2019) [1]. This trend is increasing; Zablotsky and colleagues (2019) [1] found a significant increase when they compared the two time periods of 2009–2011 (16.2%) and 2015–2017 (17.8%) in the same age group. More specifically, the percentage of children with autism spectrum disorder (ASD) increased from 1.1% to 2.5%, and the percentage of children with intellectual disabilities (ID) increased from 0.9% to 1.2% in the same age group. Considering the results of this study, the Center for Disease Control and Prevention (CDC) and the Health Resources and Services Administration (HRSA) suggested that public health and healthcare service providers plan for medical, educational, and social services to support children and their families. 

Fundamental movement skills (FMS) are required to perform daily life and sports activities and have an important influence on emotional and social development in infancy and childhood (Clark and Metcalfe, 2002; Payne and Isaacs, 2020) [2,3]. Many studies regarding the development of FMS in early childhood have shown that significant improvements in the FMS of young children may stem from educational environments as opposed to free playtime (Logan et al., 2012; Lubans et al., 2010; Morgan et al., 2013; Robinson et al., 2012) [4,5,6,7]. The level of FMS performance or the development of FMS in children has been assessed by educators and practitioners using quantitative and/or qualitative approaches (Liu et al., 2017) [8]. There are various kinds of movement assessment instruments to identify the developmental condition of FMS, such as the Test of Gross Motor Development (TGMD; Ulrich, 1985, 2000, 2019) [9,10,11], the Bruinks–Oseretsky Test of Motor Proficiency-2 (BOT-2; Bruininks, 2005) [12], and the Movement Assessment Battery for Children-2 (MABC-2; Henderson et al., 2007) [13]. The TGMD was developed to assess gross motor skill development in children with and without disabilities between 3 and 10 years of age. In 2019, the TGMD-3 was re-released with changed items and transformed performance criteria within both the locomotor and ball skill subtests (Webster and Ulrich, 2017) [14]. A total of 13 FMS skills falls under six locomotor and seven ball skills on the TGMD-3. Previous studies assessed FMS performance in children with (Brian et al., 2018; Case and Yun, 2018; Klavina et al., 2017; Staples et al., 2020) [15,16,17,18] and without disabilities (Brusseau et al., 2018; Burns et al., 2017; Kracht et al., 2020; Fu and Burns, 2018; Webster et al., 2019) [19,20,21,22,23], using the TGMD assessment instrument in educational settings to measure gross motor skills via locomotor and object control subtests. This assessment instrument is an important indicator for monitoring the levels and progress of FMS in children with developmental disabilities (Kim et al., 2012; Kirk and Rhodes, 2011; Macdonald et al., 2017) [24,25,26].

Regarding the research on adapted physical education (APE), efforts to participate in regular physical activity and to develop motor skills for children in APE programs were proven to have positive impacts in many studies (Esentürk and Yarımkaya, 2021; Hauck et al., 2020; Lai et al., 2020; Regaieg et al., 2021; Todd et al., 2021) [27,28,29,30,31]. Those studies were implemented based on treatments in health settings, organizational research, and various professional fields and supported by scientific and objective evidence, which is referred to as evidence-based practice (EBP; American Psychological Association, 2005; Williams and Glisson, 2014) [32,33]. The importance of EBP has been emphasized in the APE area (Hutzler, 2011; Bouffard and Reid, 2012; Reid et al., 2012) [34,35,36].

Children with low levels of FMS experience several barriers to participation in physical activities (Obrusnikova and Dillon, 2011; Ohrberg, 2013; Pan et al., 2011) [37,38,39], and this has a negative effect on self-confidence and motivation (Goodwin and Thurmeier, 2006) [40]. There is no doubt that providing adequate services to enhance the acquisition and development of FMS for children with DD is important (Stodden et al., 2008; Strong et al., 2005) [41,42]. However, although the TGMD has been recognized for its convenience and is a standardized test that has been used in many studies and education services (Webster and Ulrich, 2017; Brian et al., 2018; Allen et al., 2017; Valentini et al., 2017) [14,15,43,44], some researchers have pointed out limitations of the TGMD (Kim et al., 2012; Lloyd et al., 2013; Palmer and Brian, 2017; Staples and Reid, 2010) [24,45,46,47]. For example, because this instrument measures subjective judgments, results between raters may not be consistent, especially when scoring FMS among children with disabilities (Kim et al., 2012) [24]. Kim and colleagues (2012) [24,46] found that statistically inconsistent results occurred among raters in the scoring results of TGMD-2 in children with ID. Furthermore, Palmer and Brian (2017) [46] scored the 12 FMS on the TGMD-2 and presented statistically significant differences between novice and experienced raters in 10 of the skills, except for the kick a stationary ball and gallop skills. On the other hand, Maeng et al. (2017) [48] and Rintala et al. (2017) [49] each examined the intra- and inter-rater reliability of the TGMD-3 when used by experienced experts, and reported good to excellent agreement in all skill items on the assessment instrument.

Based on the importance and common performance traits of FMS among children with DD, rater training is necessary to evaluate FMS accurately in this population, especially for pre-service or novice teachers. According to Lounsbery and Coker (2008) [50], the proficiency of general physical education (GPE) teachers in skill analysis was not significantly accurate. They presented that all PE educators should learn to improve their ability to score more accurately. In doing so, the curriculum in physical education teacher education should be reformed to provide comprehensive learning experiences in skill analysis. A systematically structured rater training program and its implementation can be utilized to maximize the use of the movement instrument and tools when PE teachers score and evaluate FMS among children, especially those with DD. The purpose of this study was to investigate the impact of a TGMD-3 rater training program on pre-service APE teachers’ ability to score fundamental motor skills for children with DD. By establishing standards, we intend to develop a training program to evaluate the FMS of children with DD more accurately in the field of physical education, through analyzing the sub-technology and performance criteria of the TGMD-3. In addition, the results of this study provided evidence not only of the effectiveness of rater training, but also the development of practical rater training protocols. It will also contribute to accurately evaluating the FMS of children with disabilities, enabling the provision of effective teaching.

## 2. Materials and Methods

### 2.1. Participants

Participants were college students majoring in adapted physical education in South Korea. The inclusion criteria were as follows: (a) majoring in the APE program, (b) had not taken the motor development class in the program, and (c) did not score fundamental motor skills on the TGMD-3. The author recruited participants in two APE classes to ask for the cooperation of instructors. After explaining the experimental purpose, procedures, risks, and benefits of this study, the author obtained written informed consent from the participants. Two groups in the present study, the experimental and control groups, were randomly selected between the two classes. A priori sample size was calculated using G*Power (version 3.1.9.7; Franz Faul, Kiel, Germany) with a medium effect size (0.5) to support detected significance in the results. The sample size of this study required a minimum sample size of 38 participants with a power of 95% and an alpha of 0.05. In the experimental group, a total of 39 pre-service adapted physical education teachers (12 females, mean age ± SD = 21.73 ± 1.68) participated in the TGMD-3 training program of this study. The experimental group completed the training program and scored FMS on the TGMD for a child with developmental disabilities. For the control group, a total of 30 pre-service APE teachers (13 females, mean age ± SD = 21.15 ± 1.24) participated in the study. The control group was given general information about the TGMD-3 and the scoring methods of FMS on the assessment instrument.

### 2.2. Training Program of the TGMD-3

To develop the training program, the present study established the information of the TGMD-3, including 13 FMS with performance criteria for pre-service teachers (i.e., novices) to score FMS on the assessment instrument. This training program was also intended to train pre-service teachers on how to accurately score when considering the components of each performance criterion, as well as to understand the behavioral and movement characteristics of children with DD, in order to evaluate their performance on the TGMD-3. Regarding the components of the performance criteria, for instance, the training program clarified what “Rotate” and “Derotate” mean, as well as the difference to the two-hand striking skill on the TGMD-3. Additionally, if a child with DD shows challenging behaviors in the performance trial, such as stereotyped patterns of behavior or unexpected movements, how to correctly score according to each performance criterion was covered. First, a review of reference studies related to the TGMD-2 and 3 was conducted to arrange findings and suggestions for the psychometric properties of those instruments. Six experts in motor development and adapted physical education (APE) areas (motor development: 3; APE: 3) were engaged in deriving limitations and improvement methods when using the TGMD-3 for children with DD. This study used the administration video by Dr. Ulrich (2014) [51], who is the TGMD developer, to describe the general information of the TGMD-3, and TGMD-3 skill performance videos of children with DD in order to provide sample performances and scoring in the training program for the pre-service teachers. The rater training program consisted of an introduction to the TGMD-3, the scoring method, performance characteristics of children with DD, and the guidelines for correct scoring according to each performance criterion (see Table 1).

The training program provided not only content on children with DD and their behavior and movement characteristics, but also scoring of FMS for children with DD as a practical experience. During the scoring practice, discussion and feedback were given to provide explicit evidence and reasoning for the correct score of either 1 or 0 on each performance criterion on the TGMD-3.

The fidelity of this training program was assessed using the instrument criteria based on fidelity and measuring internal validity (McKay et al., 2018; Kaderavek and Justice, 2010) [52,53]. The training program was recorded on video for 120 min, with the participants’ consent. Three experts scored the training program plan and program implementation on the fidelity check sheet. There were three eligibility criteria to be considered an expert rater: (a) a graduate degree in motor development or adapted physical education/activity, (b) experience assessing skills on the TGMD-3, and (c) a minimum of 5 years’ experience teaching FMS curricular content to children with DD in physical education, adapted physical education, or physical activity programs. The intervention program fidelity was 93.4%, with a high inter-observer agreement of 92% on the training program plan and recorded video.

### 2.3. Study Design & Procedure

This study featured an experimental design that included a pre- and post-test, an experiment, and a control condition (see Figure 1). The dependent variables in this study were the three FMS categories (locomotor, ball skill subtest, and total skill score), and a total of 13 FMS on the TGMD-3.

The primary researcher received approval from the Institutional Review Board (IRB) for all procedures before training and data collection. First, the performance videos of a child with DD were shared with three expert raters who scored the FMS of the children with DD. The standard scores of FMS on the TGMD-3 were established by the three experts to compare with the scores of the participants. The experts independently scored the skill performance of a child with DD on the TGMD-3. Then, they had a meeting to reach 100% agreement on scores on the TGMD-3 by comparing and discussing a total of 13 skill performances with all performance criteria on the assessment instrument. 

Participants in the experiment group not only engaged in the TGMD-3 training program, but also scored 13 skills on the TGMD-3 of a child with DD two times, as a pre- and post-test. After learning general information about the TGMD-3 in the classroom setting, they watched the 13 TGMD-3 skill performances of a child with DD through a confidential web storage link and scored the skill performance on the TGMD-3 scoring form. Then, behavior and movement characteristics of a child with DD and correct scoring methods in the training program were provided to the participants before the post-test scoring. Lastly, they scored the same skill performance on the TGMD-3 again as the post-test in this study. The primary researcher collected the pre- and post-test scores of the TGMD-3 from the participants.

Participants in the control group received 20 min of introduction to the TGMD-3 in the classroom to score 13 skills on the TMGD-3 on the pre-test. After the introduction time, they were asked to score the same FMS of a child with DD as in the training program. The pre-test score sheet was submitted to the primary researcher. Considering the learning effect, the control group was asked to submit the score sheet of the same skill performance on a confidential web storage link within the next 5 days for the post-test.

### 2.4. Data Analysis

The mean and standard deviation of the skill scores in both groups were calculated for the locomotor (i.e., the sum of 6 skills) and ball skill (i.e., the sum of 7 skills) subtests, total skill scores (i.e., the sum of 13 skills), and each skill on the TGMD-3, respectively. The scoring differences between both groups (i.e., experiment and control groups) from pre- to post-scoring were also analyzed using a two-way repeated-measures analysis of variance (Two-way RM-ANOVA) to examine the improvement in scoring accuracy of pre-service teachers compared to experts. Bonferroni’s post hoc analysis was used to examine differences in the group. All analyses were performed with IBM SPSS Statistics, version 28.0 (IBM Corp, Armonk, NY, USA). The significance level of the *p*-value was below 0.05.

## 3. Results

The comparison results of locomotor, ball skill, and total gross motor skill scores between the pre-service teachers and experts are shown in Table 2. The results, based on RM-ANOVA, were for the locomotor subtest, including six skills (*F* (1, 67) = 42.52, *p* < 0.001), the ball skill subtest, including seven skills (*F* (1, 67 = 31.41, *p* < 0.001), and the total gross motor scores, consisting of 13 skills (*F* (1, 67) = 47.59, *p* < 0.001), on the TGMD-3. The scores of the locomotor subtest changed from a mean and standard deviation (*M* ± *SD*) of 32.92 ± 4.31 to 36.44 ± 2.27. The ball skill subtest scores changed from 38.56 ± 4.62 to 41.00 ± 3.66. In the total skill scores, the mean score on the pre-test (*M* = 71.49, *SD* = 7.24) was also changed on the post-test (*M* = 77.47, *SD* = 4.14). The locomotor, ball skill subtest, and total skill scores from the expert were 39.00, 46.00, and 85.00, respectively. All scores for the pre-service teachers in the experiment group were closer to the expert score compared to the control group. This indicated that the scoring of the experiment group significantly reduced in difference from the pre- to the post-test. Figure 2 shows the change of mean score differences in the locomotor, ball skill subtest, and total skill scores of the experiment and control groups between the pre- and post-test.

Each skill in the locomotor subtest on the TGMD-3 showed different results, as can be seen in Table 3. Three locomotor skills, the run, horizontal jump, and slide, were significantly improved, and were closer to the expert scores on the post-test compared to the control group. These three locomotor skills had significant *p*-values in the comparison between pre- and post-test scoring (run: *F* (1, 67) = 10.728, *p* = 0.002; horizontal jump: *F* (1, 67) = 4.409, *p* = 0.04; slide: *F* (1, 67) = 20.345, *p* = 0.000). However, other locomotor skills on the TGMD-3 (gallop, hop, and skip) did not show significant improvement (*p* > 0.05).

Regarding the ball skill subtest, the results of the comparison between pre- and post-test scoring differences compared to the expert are shown in Table 3. Among the seven ball skills on the TGMD-3, three ball skills (two-hand strike: *F* (1, 67) = 4.770, *p* = 0.032; dribble: *F* (1, 67) = 8.361, *p* = 0.005; and kick: *F* (1, 67) = 4.119, *p* = 0.046) showed significant improvement. The other four ball skills, one-hand strike, catch, overhand, and underhand throw, did not show significant improvement (*p* > 0.05).

## 4. Discussion

The purpose of this study was to investigate the effect of a rater training program for pre-service APE teachers on the scoring accuracy of fundamental motor skills (FMS) on the TGMD-3 in children with DD. Establishing standards of scoring FMS on the TGMD-3, the training program of this instrument, when used to rate children with DD, might enhance scoring accuracy on the performance criteria of the TGMD-3. The findings in this study provide evidence to satisfy the suggestions in previous studies (Kim et al., 2012; Palmer and Brian, 2017) [24,46] that emphasized the necessity for a TGMD rater training protocol in physical education teacher education programs, in order to improve scoring accuracy of FMS on the assessment instrument.

The TGMD-3 training program which was developed for the present study could be used as a basic resource for training physical education educators and pre-service teachers of children with disabilities, especially developmental disabilities. This training program could also have an initiating role in enhancing the quality of physical education and physical therapy services for children with disabilities. Though there were scoring differences on the TGMD-3 in the comparison between expert and pre-service teachers, the pre-service teachers showed several improvements in scoring on the assessment instrument following the training program. This finding is evidence for the reduction of rater effects by rater training (Weigle, 1998) [54], especially regarding the TGMD assessment instrument for children with disabilities (Kim et al., 2012) [24]. Overall scoring accuracy of the locomotor, ball skill subtests, and total gross motor skill scores on the TGMD-3 were significantly improved. According to Palmer and Brian (2017) [46], there were significant differences in both locomotor skill and object skill (i.e., the ball skill in the TGMD-3) subtests on the TGMD-2 between novice and expert raters. In the locomotor skill subtest, all skills were significantly different, with the exception of the gallop skill (*p* = 0.09). In the comparison with the present study, the scoring differences between novice (i.e., pre-service teachers) and expert raters were improved in the run, horizontal, and slide skills on the TGMD-3 compared to the control group. The other three locomotor skills (i.e., gallop, hop, skip) on the TGMD-3 did not show a significant improvement in the within-subject comparison (time × group), though several effects of the intervention in the experiment group were significant in the gallop and skip skills (*p* < 0.05). This implies that the gallop and skip skills on the TGMD-3 should be investigated to verify either the impact of the rater training program or the learning effect, according to the comprehensiveness of the performance criteria of those skills, on scoring accuracy for novices. At the same time, the content approach of the locomotor skills in the TGMD-3 rater training protocol should be improved based on differing perspectives for each performance criterion. Kim and colleagues (2012) [24] investigated rater effects in scoring FMS items on the TGMD-2 among children with intellectual disabilities, and found that the run (17.89%) and horizontal jump (12.84%) skills had relatively large error variance by rater effects compared to other locomotor skills on the instrument. The results of the present study found similar results. The mean scoring differences between the expert and novice raters in the run (experiment *M* = −1.92; control *M* = −2.90) and horizontal jump (experiment *M* = 2.36; control *M* = −2.53) skills on the pre-test were larger than in the other skills. The run, horizontal jump, and slide skills in the locomotor subtest were significantly improved according to the effect of rater training in the present study. The TGMD-3 rater training program in this study had a positive impact on reducing rater effects on the run and horizontal jump skills, as pointed out by Kim and colleagues (2012) [24], as well as on the slide skill. In addition, this implied that scoring the gallop, kick, and skip skills on the rater training program should be dealt with in more detail for each performance criterion of the skill.

For the subtest of ball skills, there was a significant improvement in the scoring accuracy for the two-hand strike, dribble, and kick skills due to the rater training intervention. The kick skill on the TGMD-2 had the lowest difference between the expert and novice raters for object control (i.e., ball skills on the TGMD-3) skills (Palmer and Brian, 2017) [46]. Additionally, Maeng et al. (2017) [48] concluded that the kick skill on the TGMD-3 had the lowest inter-rater reliability among the ball skills (ICC: 0.51). Moreover, Kim and colleagues (2012) [24] asserted that this skill had a large error variance (18.00%) by rater effects and a low agreement (ICC: 0.50). However, the present study showed that the one-hand strike skill (experiment *M* = −1.76; control *M* = −2.30) had the highest-scoring difference compared to the experts. Moreover, the two-hand strike skill showed the second highest-scoring difference (experiment *M* = −1.70; control *M* = −2.01). It could be inferred that the performance criteria of the skills on the TGMD-3 among children with DD should be examined to identify the error variances by raters according to their characteristics in future research. In other words, reliability and validity examinations of the TGMD-3 among children with DD should be considered in future studies. 

In the present study, the catch skill had the lowest scoring difference compared to the experts (both experiment and control *M* = −0.07). This implied that the catch skill has comprehensive performance criteria regardless of the expertise of raters. Interestingly, scoring the catch skill of a child with DD in the experiment group had a larger mean difference after the rater training program. The scoring difference of the control group was closer to the expert than the experimental group on the post-test. Moreover, the scoring difference of the control group compared to the expert changed from negative to positive. However, the *p*-value (*p* = 0.062) of the catch skill was close to 0.05, though it was not significant. This unusual impact of the training program may indicate that the content for correct scoring on the catch skill should aim to minimize misunderstanding of the performance criteria. 

The change in the mean differences between the two groups (i.e., experiment and control) in the overhand throw skill compared to the expert was not significant (*p* > 0.05) because the change in the scoring means from pre- to post-test was similar (experiment: 0.53; control: 0.50). The same trend of change appeared for the underhand strike skill (experiment: 0.13; control: 0.10). Regarding the one-hand strike skill, the change in the mean differences in the experiment group was not significant, although the scoring accuracy of the control group was decreased. We found that the rater training program did not provide effective training content to score the overhand throw, underhand throw, and one-hand strike skills on the TGMD-3 based on the results. These findings suggested that the TGMD-3 rater training program should describe correct scoring methods based on the proper interpretation of each performance criterion on the skills in the future rater training protocol in order to improve the scoring accuracy of novices.

Palmer and Brian (2017) [46] asserted that most skills on the TGMD-2 showed a significant scoring difference between experts and novice raters, except for the gallop in the locomotor and the kick in the object control (i.e., ball skill) subtests. Thus, they suggested training protocols and preparation to correctly score FMS on the TGMD assessment instrument. This study applied a TGMD-3 training program to examine its impact on scoring accuracy when evaluating FMS on the instrument among children with DD. There were significant impacts on the scoring accuracy of the skills on the TGMD-3, although changes in some skills did not reach statistical significance. This implies that the training program for pre-service teachers can improve scoring accuracy by reducing rater effects when evaluating FMS on the TGMD-3 among children with DD. The finding indicates that the TGMD-3 rater training program can help general and adapted physical education (GPE/APE) teachers to improve their scoring competency in skill analysis (Lounsbery and Cocker, 2008) [50], to adequately make placement decisions (Akuffo and Hodge, 2008; Columna et al., 2010) [55,56] and to prepare proper APE programs for children with disabilities (Lytle et al., 2010) [57]. 

The limitations in this study include that only one subject with DD skill performance on the TGMD-3 was scored to investigate the effect of the TGMD-3 rater training intervention on scoring accuracy. Due to varying behavior and movement characteristics of children with DD, the scoring accuracy of novices may be influenced when evaluating motor skill performance of those populations. Therefore, the results may differ in studies including children with DD who have unique types of behavior and movement characteristics. Regarding the participants, this study recruited only pre-service APE teachers who had not taken the motor development class in the program. However, the APE program curriculums are different from university to university. Thus, there is a limit to generalizing the result of this study to every pre-service APE teacher. In addition, though there was a lack of scoring accuracy among pre-service APE teachers before the training, its range was wide according to personnel experiences or their college year. Future research is recommended applying to more specific eligibility (e.g., dealing with individuals with disabilities, siblings with disabilities) for recruitment to have equivalent conditions for participants. Lastly, a TGMD-3 rater training program was executed for 120-min. There were limitations to providing information on scoring correctly not only diverse challenging behaviors of children with DD but also their various levels of skill performance on the TGMD-3. A TGMD-3 rater training program for novices should be conducted to cover more arguments in scoring FMS among children with DD in future research. 

## 5. Conclusions

This study aimed to examine the impact of a TGMD-3 rater training intervention on scoring accuracy for fundamental motor skills (FMS) among children with developmental disabilities (DD). There were significantly positive effects of the TGMD-3 rater training intervention on correctly scoring skill items on the TGMD-3 in a child with DD by pre-service adapted physical education teachers. The positive results included the following: (a) three locomotor skills (i.e., run, horizontal jump, slide); (b) three ball skills (i.e., two-hand strike, dribble, kick); (c) the locomotor subtest; (d) the ball skill subtest; and (e) total skill scores on the assessment instrument. This TGMD-3 rater training protocol can improve performance scoring by reducing the rater effects of different characteristics and experiences. It will be helpful when evaluating FMS on those measures among children with DD or a lower level of skill proficiency. However, future research involving children with DD with diverse challenging behaviors and different levels of movement performance and investigation of specific skill items and their performance criteria on the TGMD-3 should be conducted to address the limitations discussed above.

## Figures and Tables

**Figure 1 children-09-00881-f001:**
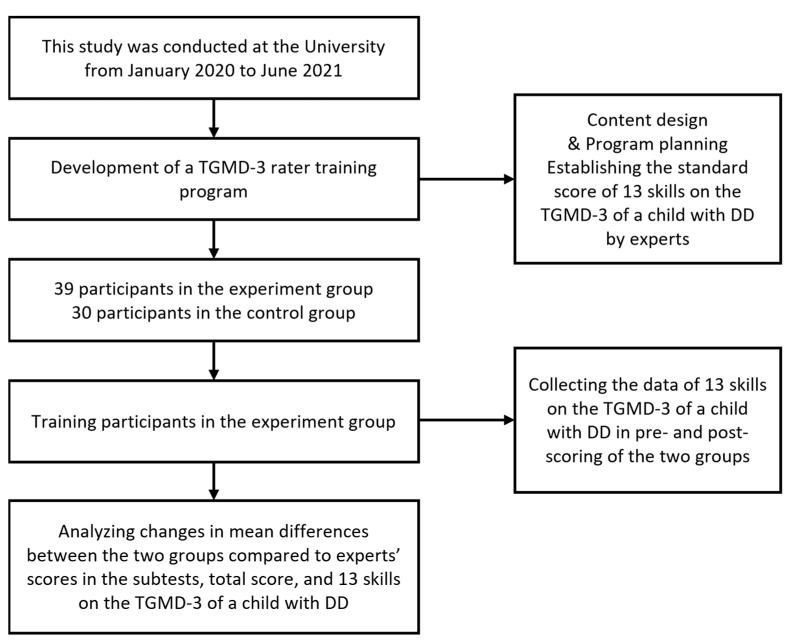
Flow chart of the study.

**Figure 2 children-09-00881-f002:**
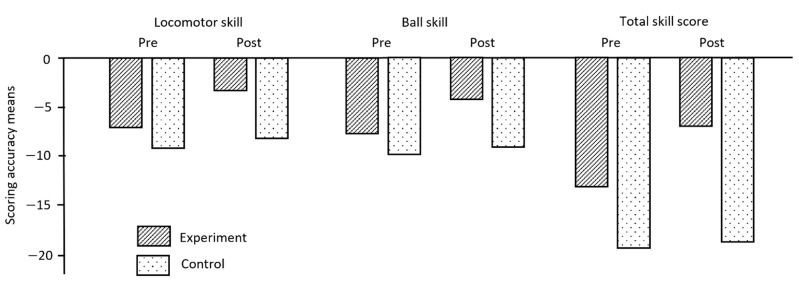
Scoring mean differences in the TGMD-3.

**Table 1 children-09-00881-t001:** Description of the TGMD-3 Rater Training Program.

Time (min)	Topic	Content	Task
20	Introduction to the TGMD-3	What is the TGMD-3 and its utilization	PPT presentation, 13 skills with performance criteria on the TGMD-3, Scoring method
		Components of the TGMD-3	
		How to score using the TGMD-3 record form	
30	Scoring practice #1	13 skills on the TGMD-3	Independent scoring
40	Introduction to developmental disabilities (DD)	What is DD	Discussion about the point of performance criteria, provide feedback on skills performance using sample videos
		Behavioral & movement characteristics of children with DD	
	Correct scoring and practical applications on the TGMD-3 in children with DD	Review of scoring 2 items on locomotor skill subtest	
		Review of scoring 2 items on ball skill subtest	
30	Scoring practice #2	13 skills on the TGMD-3	Independent scoring

TGMD-3: The Test of Gross Motor Development-Third Edition.

**Table 2 children-09-00881-t002:** Descriptive and statistical results in locomotor, ball skill, and total skill score between pre-service APE teachers and expert.

Category	Condition	Pre-Service APE Teacher ^a^		Expert	*F* (1, 67) ^b^	*p* ^b^
		Pre-Test	Post-Test			
Locomotor skills	Control	29.30 ± 3.94	30.07 ± 4.27	39.00	42.52	0.000 *
	Experiment	32.92 ± 4.31	36.44 ± 2.27			
Ball skills	Control	35.03 ± 4.87	32.27 ± 6.51	46.00	31.41	0.000 *
	Experiment	38.56 ± 4.62	41.00 ± 3.66			
Total skill score	Control	64.33 ± 8.14	65.33 ± 9.62	85.00	47.59	0.000 *
	Experiment	71.49 ± 7.24	77.47 ± 4.14			

(^a^) mean ± SD; (^b^) Test of Between-Subjects Effect (group); * *p* < 0.001.

**Table 3 children-09-00881-t003:** Descriptive and statistical results of the TGMD-3 skill score comparison between pre-service APE teachers and expert.

Category	Group	Pre-Service APE Teacher ^a^		Expert	*F* (1,67) ^b^	*p* ^b^
		Pre-Test	Post-Test			
**Locomotor skill**						
Run	Control	5.10 ± 1.52	5.53 ± 1.57	8.00	10.728	0.002 *
	Experiment	6.08 ± 1.40	7.56 ± 0.85			
Gallop	Control	4.83 ± 0.99	5.03 ± 1.30	6.00	2.681	0.106
	Experiment	5.33 ± 1.34	6.08 ± 0.70			
Hop	Control	4.70 ± 1.84	4.67 ± 1.77	5.00	0.029	0.864
	Experiment	4.92 ± 1.49	4.95 ± 0.89			
Skip	Control	3.27 ± 1.26	3.27 ± 1.26	4.00	1.822	0.182
	Experiment	4.33 ± 1.24	3.97 ± 0.71			
Horizontal jump	Control	5.47 ± 1.59	5.67 ± 1.52	8.00	4.409	0.040 *
	Experiment	5.64 ± 1.25	6.46 ± 0.91			
Slide	Control	5.93 ± 1.02	5.90 ± 0.92	8.00	20.345	0.000 *
	Experiment	6.62 ± 0.71	7.41 ± 0.60			
**Ball skill**						
Two-hand strike	Control	6.93 ± 1.36	6.90 ± 1.79	9.00	4.770	0.032 *
	Experiment	7.51 ± 1.39	8.18 ± 0.89			
One-hand strike	Control	4.70± 1.60	4.30 ± 1.86	7.00	1.639	0.205
	Experiment	5.13 ± 1.69	5.15 ± 1.35			
Dribble	Control	4.00 ± 1.44	3.87 ± 1.89	6.00	8.361	0.005 *
	Experiment	4.64 ± 1.09	5.44 ± 0.82			
Catch	Control	3.93 ± 1.31	4.07 ± 1.23	4.00	3.589	0.062
	Experiment	3.90 ± 1.21	3.56 ± 0.94			
Kick	Control	4.80 ± 1.03	4.87 ± 1.22	6.00	4.119	0.046 *
	Experiment	5.21 ± 1.42	5.82 ± 0.82			
Overhand throw	Control	5.90 ± 1.37	6.40 ± 1.52	8.00	0.017	0.897
	Experiment	7.00 ± 1.30	7.54 ± 0.85			
Underhand throw	Control	4.77 ± 1.31	4.87 ± 1.41	6.00	0.001	0.980
	Experiment	5.18 ± 1.14	5.31 ± 0.89			

(^a^) mean ± SD; (^b^) Test of Within-Subjects Effect (time × group); * *p* < 0.05.

## Data Availability

The data presented in this study are available on request from the corresponding author. The data are not publicly available due to privacy.

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
