# Peer review of "The Impact of a Rater Training Program on the TGMD-3 Scoring Accuracy of Pre-Service Adapted Physical Education Teachers"

_children, 2022, doi:10.3390/children9060881_

Round 1

Reviewer 1 Report

First of all, I would like to congratulate the authors for the article "Can the rater training program for pre-service adapted physical education teachers provide to improve the scoring accuracy of fundamental motor skills on the TGMD-3 among children with developmental disabilities?".

An interesting topic is studied, such as the analysis of fundamental motor skills in children with developmental disabilities. However, the manuscript needs to be improved to increase its quality.

Title: It is recommended to retitle the manuscript, since the proposed title is too long.

Abstract: It is very long, according to the regulations of the magazine. Requires review.

Introduction: 

You need to organize the information. This section begins by talking about developmental disabilities, then goes on to Fundamental movement skills, then to adapted physical education, and finally to FMS for children with DD.

It is recommended to delete the following paragraph (L97-100): "If there is a program or tool that is well utilized, it is considered that the effect can be maximized to use properly for its purpose. In addition, the results of this study were convinced to provide evidence of not only the effectiveness of rater training but also the development of practical rater training protocols."

Materials and Methods:

This section should describe the criteria by which the measurements were made, such as the place (ward), the clothing in which the minors took the tests, etc.

Subsection 2.4. Data analysis should be reviewed as it is not well understood.

Results:

It is ok.

Discussion:

It is ok.

Conclusions:

It is ok.

References:

References should be checked throughout the manuscript as they are entered in two different formats. Correct.

It is necessary to review the entire text since there are paragraphs that are not understood. Check.

Taking these aspects into account, it is considered that the quality of the manuscript will improve to be published by the journal. Many thanks.

Author Response

We really appreciate your valuable review.

Thank you,

Reviewer 2 Report

Dear Authors 

I am glad I have been given a chance to review your study and I hope my comments will help to enhance the article. 

Title and abstract as well as key words are in line with the text, informative and sufficient for a reader to get the idea what the study was about. 

Introduction - here I think you need to expand a little bit and work more on strengthening the rationale of the study. In brief, you are trying to convince the academic world that when students are better trained they can rate children more efficiently during they assessment in PE teaching and learning process, but this should be true in all the cases - better training better skills, no matter what skills (also including didactic skills). So, where the novelty of your study? Maybe in the Introduction you should dig deeper for some references in for example Journals concerning Measurements and evaluation in education or look into "Developing Skill-Analysis Competency in Physical Education Teachers, Quest" or look for reference to those modern techniques of assessment like in the "Reliability of pre-service physical education teachers' coding of teaching videos using software ect.  

You can (and should) even present a broader picture of the situation - maybe those Major students of PE are not ready (generally not well trained) in the area of evaluation skills (look into "Are Australian pre-service physical education teachers prepared to teach inclusive physical education?). 

Methods - this section is well written, it gives a reader a sense of research methodology used. Tools are defined and described with details. Although sample sizes of the research groups are not impressive and you did not provide any information on how the groups were selected (please do!), I think you also need to give a brief description of what was the level of competencies in assessment of motor skills - not just concerning children with DD, but all children and whether the students had already had any experience in working with DD children before joining your study. 

Statistical methods, although a bit sophisticated as for that small groups but have been used rightfully. 

Results are presented in a clean, neat manner and a reader can get a good understanding of scoring by both groups. 

Discussion - here again you need to be more critical on your findings. It is obvious that after 120 minutes of training there will be some improvements, but it takes years to get some expertise in the field - in this case in this specific part of the teaching process - evaluation and assessment accuracy.

The tempo of gaining better accuracy will depend on many factors - exposure for such situations, helpful guidance of someone more experience, own self-development, better understanding and more emotional conciousness in working with DD, teaching standards and expectations, of self-critical skills in teacher's professional development, ect. 

You should make your points at least to some of those above mentioned factors. You can look into "Pro-service physical education teachers' implementation of TGfU tennis: Assessing elementary students' game play using the GPA". You can also see how cultural education can play a role in pre-service teachers accuracy of self-evaluation of didactic competencies - look into "Comperative study on self-assessment of teaching competencies of PE student teachers from Poland and Kosovo. And another paper that may be of interest for you "Preparing pre-service primary school teachers to assess fundamental motor skills: two skills and two approaches.  

I think inclusion of more factors into your discussion will make you text more interesting for a broader spectrum of readers (minding that the sample size of the research groups were not that big and conclusions should be treated with cautions). 

References can be expanded as well.  

Author Response

(The authors gave the same response as above.)

Reviewer 3 Report

Dear authors, the article entitled "Can the rater training program for pre-service adapted physical education teachers provide to improve the scoring accuracy of 3fundamental motor skills on the TGMD-3 among children with developmental disabilities? " has been reviewed.

I have enjoyed your work very much. I consider it to be close to being published in its current format. The data obtained are truly astonishing.

The theoretical framework is well argued. The gap in previous research is detected.

The methodology is adequate, as well as the statistical tests carried out, obtaining quite striking results.

Finally, maybe it is because of the novelty of your work, but it would be interesting to reinforce the discussion and that it is ordered according to the objectives of your research.

Finally, try to connect the limitations of your work with the prospective studies that you consider.

Congratulations on your work

Author Response

(The authors gave the same response as above.)

Round 2

Reviewer 2 Report

I can see that you have considered my comments and suggestions and included most of them into the text. This has improved your presentation a little bit.